# Workforce Career Development in Public Health, Health Education, and the Health Services: Insights from 30 Years of Cross-Disciplinary National and International Mentoring

**DOI:** 10.3390/ijerph22050729

**Published:** 2025-05-02

**Authors:** Holly Blake

**Affiliations:** 1School of Health Sciences, Faculty of Medicine and Health Sciences, University of Nottingham, Nottingham NG7 2HA, UK; holly.blake@nottingham.ac.uk; 2NIHR Nottingham Biomedical Research Centre, Nottingham NG7 2UH, UK

**Keywords:** mentoring, careers, professional development, workforce, higher education

## Abstract

This paper presents my personal experiences of cross-disciplinary national and international academic mentoring over 30 years in a higher education setting, supporting 605 mentees in public health, health education, and the health services. I supported mentees at diverse career stages through (a) one-to-one mentoring relationships (*n* = 231 mentees; from the UK, Europe, Middle East, Africa, Asia, and Australasia; academics, industry, and healthcare professionals), (b) a cross-faculty structured mentoring programme (*n* = 52; junior faculty), (c) a cross-institutional interprofessional internship programme including mentoring and public health placements (*n* = 302 interns; from five universities), and (d) an interprofessional learning programme in workplace health called WHIRL, which was embedded within Test@Work, a public health innovation bridging health promotion practice, research, and industry and involving peer mentoring and mentor support (*n* = 20; volunteer healthcare trainees). In this paper, I outline the broader concept of mentoring, together with an overview of mentoring types, uses, and benefits. The diverse contexts in which mentoring occurs are discussed: (i) micro-mentoring, (ii) inducting new staff, (iii) peer mentoring, (iv) career transition moments, (v) career advancement mentoring, (vi) diversity mentoring, (vii) knowledge sharing mentoring, (viii) collaborative learning and support mentoring, and (ix) leadership development mentoring. The challenges of mentoring are presented alongside suggested actions to take. I advocate for mentoring evaluation and provide a worked example of measuring the outcomes of one-to-one mentoring using The Career Support and Psychosocial Support Scales (online survey; *n* = 103 mentees; from 22 countries). Finally, I reflect upon the diversity of mentoring experiences, with activities and benefits categorised into six key areas: interpersonal relationships; networking opportunities; enhancing knowledge and skills; employment, reward and recognition; support for under-served groups; and convening communities of practice.

## 1. Introduction

There are many definitions of mentoring [1], although here, The European Mentoring and Coaching Council (EMCC Global) definition is adopted: “Mentoring is a learning relationship, involving the sharing of skills, knowledge, and expertise between a mentor and mentee through developmental conversations, experience sharing, and role modelling. The relationship may cover a wide variety of contexts and is an inclusive two-way partnership for mutual learning that values differences [2]”.

Studies of mentoring in the higher education setting have largely been conducted in the United States, with few in the United Kingdom (UK), and they tend to focus only on the mentoring of college or graduate students, rather than faculty [1,3]. Nonetheless, in higher education contexts, it is widely accepted that mentorship plays a clear role in staff career development and academic success (Figure 1). Mentoring is associated with retention in academia [4], career satisfaction [5], self-efficacy [6,7] and expanded professional networks [8]. Robust mentorship is identified as a critical factor in facilitating health and social care professionals to transition from clinical practice into academia [9]. More broadly, mentoring can directly improve the outcomes often measured by key performance indicators at an organisational level. It can improve the quality of teaching, and can ultimately improve student learning [10,11], increase the likelihood of individuals submitting new grants and obtaining funding [12,13], and lead to more scientific publications [12,14,15]. In the context of increasing financial challenges and budgetary constraints in higher education [16,17], the vital contribution of mentoring towards meeting institutional outcomes should be recognised. Taking the enhanced quality of teaching as one example, it is postulated that improved instructional quality can impact net revenue [18]. Yet, institutions rarely appear to consider mentoring within the context of financial investment in the workforce and the ‘return-on-investment’ that could be achieved through a focus on the growth and development of existing faculty [19].

Mentoring also plays a key role in supporting career development for minoritised groups, as an inclusive practice that can assist in ‘levelling the playing field’ for those with barriers to progression (e.g., race: [20]; disability: [21]). However, for those in receipt of mentoring, the quality of mentoring experiences can be variable [22], and a substantial proportion of early- and mid-career faculty have no access to mentoring, either through informal mentoring (independently sourced relationship) or through access to formal mentoring programmes (access to an official scheme within an organisation) [23]. Mentoring in higher education is recognised to have high value, and many organisations advocate for mentoring as a relevant ‘citizenship’ activity. However, it is voluntary, rarely incentivised, and often not considered to be equal in value to other core activities such as teaching, research, and academic service roles. This may impact individuals’ willingness to engage in mentoring and their level of commitment to it, despite the clear link with academic success for staff recipients.

The overall aim of this paper is to share my personal experiences of mentoring over 30 years to share mentoring examples, and encourage mentoring engagement among academic faculty. The objectives are to (a) provide an overview of mentoring types, uses, and benefits; (b) share the diverse contexts in which my mentoring has occurred, (c) identify the challenges of mentoring and the suggested actions to prevent or manage problems; (d) advocate for mentoring evaluation and provide a worked example of a formal evaluation approach; and (e) categorise six key areas of focus for a positive mentoring experience.

## 2. Methods

### 2.1. Professional Frameworks and Standards

My own mentoring practice has been influenced by the tools and techniques described by Lancer and colleagues [24], who provide invaluable resources for mentors looking to enhance their professional practice, across diverse areas such as preparation for mentoring, rapport building and contracting, managing relationships, and dealing with setbacks. Naturally, my approach to mentoring has evolved over the years with the increasing availability of professional practice frameworks, toolkits, checklists, and guidelines. EMMC Global, for example, describe four distinct ‘indicative’ levels of professional expertise (foundation, practitioner, senior practitioner, and master practitioner) for mentoring, coaching, and leadership [2]. Skills at each level align with eight core standards which are useful to guide progression through the mentoring process (Figure 2).

Many professional bodies in the UK provide useful checklists, factsheets, and guidance to support the mentoring process (e.g., Chartered Management Institute [25]; Chartered Institute of Personnel and Development [26]).

### 2.2. Mentoring Recipients and Contexts

From my position within a UK higher education setting, I have mentored across the public, private and third sector. I have primarily mentored researchers, academics, healthcare professionals, and managers or senior leaders, mostly working within higher education or the health services, with a minority of mentees employed in industry settings. My mentees have been based in the UK, Europe, Middle East, Africa, Asia, and Australasia. The professional disciplines of my mentees have included (but are not limited to) psychology (health, occupational, clinical), public health, medicine, behavioural science, advanced clinical practice, nursing, midwifery, physiotherapy, sports science, occupational therapy, occupational health, radiography, biomedical sciences, veterinary medicine, and other non-clinical professions associated with health services (e.g., human resources, health informatics, computer sciences, business and management).

In Table 1, I present the contexts in which I have provided mentoring, the suggested ways in which the approach could be used (‘use cases’), and the benefits of each approach. These include (i) micro-mentoring, (ii) inducting new staff, (iii) peer mentoring, (iv) career transition moments, (v) career advancement mentoring, (vi) diversity mentoring, (vii) knowledge sharing mentoring, (viii) collaborative learning and support mentoring, and (ix) leadership development mentoring.

#### 2.2.1. One-to-One Mentoring Relationships

While I have engaged in all mentoring types described above, my mentoring journey started in the 1990s when I acted as a disability mentor playing a key role in fostering confidence and developing skills in a disabled staff member as well as supporting their access to professional development (“diversity mentoring”). Subsequently, I have mentored a total of 231 individuals from the UK, Europe, Middle East, Africa, Asia, and Australasia in one-to-one relationships. These mentees were academics, industry and/or healthcare professionals. The mentoring primarily focused on “career advancement” or “diversity” and involved the provision of support to individuals with barriers to progression based on certain characteristics (e.g., gender, disability, socioeconomic status, ethnicity). These relationships were initiated through direct contact or were matched through a structured mentoring programme associated with an academic institution or professional body.

#### 2.2.2. Establishing and Leading Structured Programmes

I established and directed three structured programmes which have provided mentoring for a further 374 individuals.
(a)Cross-faculty structured mentoring programme

First, I established and led a structured mentoring programme in a higher education setting, which supported 52 early career researchers and junior faculty. The stages and processes I followed to develop and evaluate the programme are outlined in Table 2.
(b)Cross-institutional interprofessional internship programme

Next, I established and directed a cross-institutional interprofessional internship programme, which created 302 internships for early career researchers from across the health and social sciences, and from five different universities. The internships were focused on capacity building and exposure to the ‘real-world’ experiences of public health initiatives and workplace health promotion services. In this scheme, interns received structured training sessions in research and evaluation methods (which I delivered), then undertook supervised placements in industry or health service settings (supervised by me as an academic mentor and working with a placement mentor). This involved group mentoring and skills-building, individual check-ins, and feedback sessions with both mentors (Figure 3). I maintained one-to-one mentoring relationships with many of the interns after the programme ended to support their transition into employment in academia, healthcare, or industry, and their career development thereon.
(c)Interprofessional learning programme: ‘WHIRL in Test@Work’

The third structured programme I established and led was an interprofessional learning programme called ‘WHIRL: Workplace Health Interprofessional Learning’ [27]. WHIRL was embedded within a research study involving a public health intervention called Test@Work [27,28,29]. Test@Work involved the delivery of multicomponent health checks, including voluntary HIV testing to workers in the construction industry. WHIRL [27] was a sub-project within Test@Work which created development opportunities for 20 volunteers (healthcare trainees in medicine, *n* = 1, nursing, *n* = 15, physiotherapy, *n* = 2, and health psychology, *n* = 2), who worked under supervision in multi-professional teams to deliver general health checks. The health checks included confidential measures of weight and height, the calculation of body mass index (BMI), waist and waist-to-hip ratio measurements, blood pressure, and a screening test for mental wellbeing. The volunteers conducted the health checks (except for the HIV testing) and gave participants their results together with some brief health advice and signposting to health and wellbeing services where relevant. Volunteers delivered general health checks to 464 construction workers at 21 health check events. The volunteers undertook a four-part training and support package of trainer-led education (co-delivered by myself, nursing, and medical colleagues), observations of practice (observing nursing and medical colleagues), self-directed learning (with a package of materials developed by the Test@Work team), and clinical supervision (delivered by a nurse colleague), together with peer mentoring (from more experienced volunteers within the programme) and mentor support (from myself as a lead for both the WHIRL and Test@Work programmes). Full details are published elsewhere [27,28,29,30] and the purposes of peer mentoring and peer support, together with their outcomes, are shown in Figure 4.

### 2.3. Evaluating Process and Outcomes

My approach to the assessment of mentoring process and outcomes has been more or less formal depending on the nature of the relationship and individuals’ personal development needs. This ranged from brief conversations and ‘check-ins’ to confirm the appropriateness of the process (e.g., clarity of purpose, frequency and duration of meetings, face-to-face versus online meetings) through to more structured approaches to evaluate outcomes, that is, the extent to which learning and change had occurred (e.g., in relation to behaviours, performance, leadership, self-esteem, personal confidence or resilience, and wellbeing). For the latter, I used reflective models, 360° feedback, and evaluation surveys.

I mapped a process-oriented model of mentorship [31] onto my assessment methodologies to measure the active functions of mentorship such as career support (i.e., sponsorship, exposure and visibility, challenging assignments or projects, and protection) and psychosocial support (i.e., acceptance, encouragement, confidence-building, friendship). This included more passive functions, such as role modelling, in which a mentor acts as an example of the norms, attitudes, and behaviours required to achieve success [32,33]. A simplified process-oriented model of mentorship is shown in Figure 5.

#### 2.3.1. Reflection in Mentoring

Reflection is an important part of the mentoring process and can be used in two ways:(1)To help the mentee to reflect on thoughts, processes, and experiences (e.g., to assist them in developing new knowledge, considering alternative perspectives, finding solutions, and ensuring mentoring is meeting their needs). This might involve working through a reflective model with a mentee. There are many models of reflection that could be used. One example that I have frequently used is the Gibbs Reflective Model [34], which offers a framework for examining experiences through six stages: description of the experience; feelings and thoughts about the experience; evaluation of the experience (good and bad); analysis to make sense of the situation; concluding with what you learned and what you could have done differently; and creating an action plan for how you would deal with similar situations in the future, or general changes you might find appropriate. Alternatively, reflective questions can simply be added to the end of each mentoring session, such as ‘What has been a success or challenge since we last spoke? What can we learn from that?’.(2)To reflect on one’s own practice as a mentor, recognising that we are imperfect and influenced by our own experiences, thoughts, biases, and way of doing things. Again, a reflective model could be used to consider one’s own role as a mentor, how effective one is at mentoring and role modelling, and how one’s positionality and privilege can contribute to, or influence, the relationship. I initially engaged in reflective practice on an individual level. This was largely due to a lack of obvious alternative approaches, particularly in the earlier years of my career, when fewer colleagues were engaging in, or reflecting on, mentoring practice. In more recent years, mentoring has become more prominent in career development discussions, and in some settings, best practice involves the establishment of communities of practice through which academic staff can discuss mentoring practices and access peer-to-peer support and/or supervision. I would now advocate a “supervised/supported” approach, since the supervision of mentoring practice not only supports the professional development of those acting as mentors, but can provide a quality assurance mechanism, helping to maintain high standards in mentoring by providing oversight and guidance.

#### 2.3.2. Seeking 360° Feedback

Evaluation is key to establishing whether the mentorship is achieving the desired goals and outcomes. Seeking ‘360° feedback’ is a useful process in which confidential, anonymous evaluations are sought from the people who work around us. In the context of employment, this typically includes managers or supervisors, peers, and direct reports, but it can also be used in the context of mentoring as a process of leadership development [35]. Such feedback is usually solicited via online survey, including specific questions that enquire about a persons’ attributes and competencies, not solely about their performance. This might include closed- and open-ended questions relating to leadership abilities, strategic mindset, communication, interpersonal skills, decision-making, problem solving, motivation for work and their own role, and efficiency (relating to how the individual reacts to the organisation’s goals and values). While an individual may nominate those from which they wish to request feedback and distribute the survey link, the responses are often collated by a third party, with feedback provided to individuals in summarised (and non-identifiable) form. My own experience of 360° feedback has been in structured leadership programmes (as an attendee), and in a structured mentoring programme (as a mentee), and provided highly valuable information about the views of those with whom I worked closely. However, it is important to note the benefits and potential caveats of 360° feedback. If respondents do not work closely with the person seeking feedback, their comments may be less informative. Further, the environment in which we work can play a significant role in the culture and norms related to the provision of feedback and how it is accepted and acted upon. It has been shown that leaders who work in a favourable feedback environment prior to undergoing a process of 360° feedback have better outcomes (e.g., improved leader performance) than those who work in an unfavourable feedback environment [36]. Throughout all of my mentoring roles, I have had the privilege of working in a higher education setting, largely in healthcare education, which is a favourable feedback environment in which soliciting and using feedback (e.g., staff–staff, staff–student, student–student) is actively welcomed and highly normalised [37]. Therefore, the process of soliciting 360° feedback as a mentor in higher education (staff–staff) is an appropriate and arguably an anticipated approach.

#### 2.3.3. Evaluation Surveys

I am a strong advocate of evaluating practice. Recently, I used a mixed methods approach to mentoring evaluation, including a quantitative rating for mentoring support and qualitative feedback. These data are, by nature, collected and interpreted by the mentor (rather than by an independent evaluator), as the results have direct relation to an individual’s mentoring behaviour and performance, and to potential avenues for future development. Here, I present a worked example of how I formally evaluated the outcomes of my mentoring using a questionnaire survey.

Referring to Table 1 (mentoring types), the 605 mentees I have supported does not include the vast numbers of individuals for whom I have provided very short-term or single-session ‘micro-mentoring’, ‘peer-to-peer’ mentoring, those who I have ‘inducted/onboarded’ into new job roles, or those for whom I have only provided ‘knowledge sharing’. Academics provide this support as a routine part of their day-to-day job role and/or academic service. Except for inductions, these regular transactions are rarely documented, as they are so integral to an academics’ role. Rather, I sent the evaluation survey to individuals for whom I had provided individual mentoring support spanning several months (at minimum) or many years (up to 10+) and focused on career advancement or transition, diversity, collaborative learning, and leadership development.

I distributed an online survey, hosted on Microsoft Forms, to mentees which remained open between 1 July 2024 and 28 December 2024. Although I was aware of the individuals to whom the survey was sent, the survey could be completed anonymously. It had 18 items. There were five demographic items, including gender identification, age, ethnicity, current and main occupation, and home country; and three items related to mentoring characteristics, including the year mentoring support was first provided, duration of mentoring support (in years), and a free-text item relating to the perceived benefits of mentoring support. Two primary functions have been identified in mentoring, including the provision of psychosocial support (which includes role modelling) and offering career or instrumental support (which includes providing challenging work that facilitates skill development) [38]. To measure the outcomes of these functions, the Career Support and Psychosocial Support Scales [39] comprised the remaining 10 items. These scales have been validated in a higher education context [40]. The Career Support Scale has five items, on which respondents rate to what extent (1) “did I provide you with opportunities that stretched you professionally?”, (2) “did I create opportunities for visibility for you in your career?”, (3) “did I open doors for you professionally?”, (4) “did I act as a sponsor (supporter) of you?”, and (5) “did I act as a buffer for you from situations that could threaten your career achievement?”. The Psychosocial Support Scale has five items on which respondents rate to what extent (1) “did I care and share in ways that extended beyond the requirements of work?”, (2) “did I counsel you on (talk to you about) non-work-related issues?”, (3) “did I offer you support, respect, and encouragement?”, (4) “do you consider me a friend?”, and (5) “do I confirm and affirm your identify and sense of self?” The response options for both scales ranged from 1 (never, not at all) to 7 (to the maximum extent possible). The item responses are summed to measure either career or psychosocial support, with scores ranging from 5 to 35 for each scale, and higher scores indicating greater perceived support.

## 3. Results

### 3.1. Learning from Diverse Mentoring Approaches

My experiences of mentoring through the approaches shown in Table 1 uncovered numerous challenges. In my case, these largely related to failures in contracting and this became less frequent over time. My ability to anticipate and manage challenges grew through a process of (i) observing my own mentors’ practices; (ii) undertaking mentor training (through a faculty-delivered structured mentoring programme including pre-course material and a webinar); (iii) undertaking other training with transferable learning, including two year-long leadership programmes (run by my institution, including senior leaders and research leaders training), and professional qualifications in both coaching and management (accredited with a national body); and (iv) increasing my numbers of mentees over the years, meaning I became a more experienced and adaptable mentor over time.

The greatest challenge has been enabling mentoring closure. Mentees with whom I have developed relationships with over many years still approach me for support at every decision-making juncture in their careers. While helping people to reach decisions or find solutions is arguably even more rewarding in long-established mentee/mentor relationships, it can also be exceptionally time-consuming for the mentor, who may be supporting multiple mentees at any one time. This is naturally dependant on the nature of the mentoring relationship, since structured programmes often have a distinct final mentoring ‘phase’ that is time-limited and agreed at the outset, while independently organised mentoring dyads can be flexible and transitional, with relationships extending beyond formal mentoring to an informal post-mentoring professional relationship where support and guidance may continue to be provided. Others have identified that mentees may maintain such ‘mentorship interactions’ with former mentors over time [23]. Overall, the challenges I have identified in the mentoring processes are outlined in Table 3, together with the actions I have taken to prevent or manage such situations.

In the early years of my mentoring practice, there was little training available on the process, but training is now much more accessible. Mentor training has implications for the quality of mentoring and mentoring outcomes, and can potentially address all challenges and associated actions listed above. The types of training and support vary considerably in content, scope, and depth. Some training involves the simple provision of materials (such as documents or slide sets), or single-session events (such as workshops or webinars). Mentor training might be included within a teaching certificate or recognition award. In the UK for example, mentoring could be supported and recognised within a fellowship awarded by AdvanceHE. Other training could be provided through formal peer mentoring programmes or ‘communities of practice’ established within academic staff pools. Structured mentor training programmes might include tools and frameworks for staff developers which focus on empowering mentees and fostering a culture of learning. Such training can emphasise the strategic benefits of embedding mentoring in higher education, and may cover mentoring theories, critical reflection, problem solving, communication skills, adaptability, and practical implementation.

### 3.2. Formal Evaluation Results

Of the 231 mentees I supported between 1997 and 2024 (outside of the structured programmes I established myself, and those in receipt of one-to-one mentoring), there were 126 (54.5%) individuals that were contactable at the time of the survey. Of these, 103 (82%) responded. Respondents identified as women (*n* = 75), men (*n* = 22), non-binary (*n* = 4), or preferred not to say (*n* = 2). They were aged under 18 (*n* = 3), 18–24 (*n* = 25), 25–34 (*n* = 27), 35–44 (*n* = 34), 45–54 (*n* = 8), or 55–64 (*n* = 6) years. Respondents described their ethnicity as Asian or Asian British (*n* = 23); Black, Black British, Caribbean, or African (*n* = 13); Mixed or multiple ethnic groups (*n* = 5); White (*n* = 40); or Arab (*n* = 22). Their reported current and main occupations included the following: higher education academic/teacher (*n* = 21), early career researcher (*n* = 41: 11 health professionals undertaking doctoral research; 27 fixed-term researchers working in a higher education setting; 4 permanent researchers working in an industry setting), nurse/midwife (*n* = 22), allied health professional (*n* = 9), manager (*n* = 6), or other (*n* = 3). Many had dual roles; the academics and researchers had diverse subject disciplines, and many were also registered healthcare professionals. All the doctoral researchers were (at the time of mentoring) employed in higher education settings (e.g., as academic lecturers) or clinical roles in the health services (e.g., registered nurses) either in the UK or abroad and were released for study, aspiring to have a clinical academic career. The respondents came from 22 countries, including England, Ireland, Scotland, Wales, Greece, Saudi Arabia, Jordan, Oman, Iraq, Iran, Ghana, Nigeria, Malawi, Barbados, Thailand, Indonesia, Malaysia, Sri Lanka, China, Taiwan, Pakistan, and India.

Mentoring took place between 1997 and 2024. The respondents received one-off mentoring support comprising a series of sessions over several months (*n* = 12), or individual mentoring for one year (*n* = 17), 2–4 years (*n* = 32), 5–7 years (*n* = 15), 8–10 years (*n* = 7), or over 10 years (*n* = 4). The mentoring frequency was weekly, monthly, or three-monthly depending on the nature of the mentoring relationship and context. The results of my mentoring survey are shown in Table 4.

The five-item versions of the Career Support and Psychosocial Support Scales were used here. The overall mean scores were very high for each item, which indicated that career and psychological support was successfully achieved through my mentoring relationships. I reflected on three items on which a minority of mentees gave lower scores. Some mentees gave lower scores on the career support item “Acts as a buffer for you from situations that could threaten your career achievement”. It is worth noting that a study of the psychometric properties of the Career Support and Psychosocial Support Scales identified a better model fit when this item was deleted from the Career Support Scale [40], since not all respondents (i.e., mentees) will have experienced a threatening event that required such a response from someone in their developmental network (i.e., mentor). Two other items received lower scores from a minority of international mentees: “Acts as a sponsor of you” and “is a friend of yours”. This could simply represent a low rating in these areas. However, ‘sponsor’ in this context refers to ‘supporter’ and it became clear that some international mentees had interpreted ‘sponsor’ to mean ‘provider of financial support’, e.g., a fellowship grant. For those mentees, their ‘sponsor’ was therefore a funder or employer, usually in their home country, and therefore they did not perceive ‘sponsorship’ to be associated with the mentoring relationship. Regarding ‘friendship’, the lowest responses to this item came from mentees for whom the relationship was much shorter in duration (meaning mentoring may not have had time to transition to ‘friendship’), or mentees from cultures in which hierarchical structures are deeply embedded, meaning that mentors are more likely to be perceived to have supervisory authority when providing guidance, rather than being ‘friends’. This perhaps suggests a need to validate the measures in different populations.

Qualitative comments from free-text responses were analysed thematically to identify the diverse ways in which mentoring was perceived to benefit recipients. These benefits and mentoring activities were categorised into six key themes shown in Figure 6: interpersonal relationships; networking opportunities; enhancing knowledge and skills; employment, reward, and recognition; support for under-served groups; and convening communities of practice. All mentees had received mentoring support spanning multiple areas.

The evaluation also highlighted the specific areas where my mentoring had led to improvements across public health, health education and/or the health services. Mentees reported having actively engaged in public health or health services research, audits, and/or quality improvement projects. Much of this activity had led to publication or grant applications. This was widely reported across the pool of respondents, but for certain healthcare professionals, it was a particularly significant aspect of their development. National studies show that advanced clinical practitioners (ACPs), for example, struggle to commit time to work across the four pillars of advanced practice, particularly the research pillar [41], and the need for mentorship has previously been recognised [42]. Many of the mentees were ACPs from diverse professions, who benefited from support with navigating research engagement alongside their busy clinical roles and managerial responsibilities and onwards sharing of those insights with their teams. Mentees also included nurses, midwives, and allied health professionals at various stages of the clinical academic career pathway who went on to secure academic internships or fellowships. A minority of mentees who were healthcare professionals working in industry settings referred to the value of sustaining a professional connection with higher education and healthcare environments.

Some mentees described the attainment of new knowledge in specific health-related fields (e.g., public health and health promotion, behaviour change techniques, patient self-management approaches, work and health, digital innovations in health). Those who were engaged in research reported positive impacts from the research interventions they had either been involved in, or led, on patient populations in the UK and abroad, mostly in hospital and community settings. Some mentees went on to win local, regional, national, or international prizes and awards for their work and practices (e.g., teaching certificates and awards, recognition or service awards from national or international societies or professional bodies, research funding awards, prizes for disseminated work such as ‘best paper’, research impact awards). Many reported increased engagement in research dissemination activities, such as conferences, masterclasses, seminars, or webinars.

Mentees working in higher education reported enhanced approaches to health education and increased engagement in health education scholarship activity, such as educational conferences, interprofessional learning projects, or writing and editing textbooks. Mentees reported greater involvement in citizenship activities, such as reviewing papers for scientific journals, reviewing books for academic publishers, becoming a member of research funding panels, supporting wider agendas within healthcare or higher education (e.g., contributing to activities to support the Teaching Excellence Framework (TEF) and Research Excellence Framework (REF)), and contributing to agendas for staff wellbeing; equality, diversity, and inclusion; global engagement; digital; and civic and regional engagement. Mentees had joined local, regional, or national boards at various institutions, joined government Think Tanks, worked with regional public health teams, and delivered health-related projects with local communities.

There were reports of increased engagement in patient and public involvement activities (and associated training), which is a vital aspect of health education, health research, and health service improvement. Some mentees reported having developed new clinical skills or transitioned to new clinical areas either in healthcare or within industry. Mentees described the acquisition of new leadership skills, and/or improved multi- and inter-disciplinary teamworking across professions and sectors. They reported an enhanced ability to manage competing demands, a challenge raised many times by those attempting to balance clinical responsibilities with academic activity (teaching, research, management, and broader citizenship). The COVID-19 pandemic (March 2020–May 2023) brought significant changes to the world of work, and the mentoring process helped many to navigate the practical and emotional challenges of job role changes, or a transition to remote or hybrid work patterns. Some mentees reported significant benefits of mentoring for their mental wellbeing. Mentees often described how their work-related confidence had increased due to the mentoring, which for some, ultimately led to better patient care. This was associated with their increased knowledge from seeking training and development opportunities, and improved communication within their teams and with patients and their families. New knowledge, skills, confidence, and/or qualifications had resulted in many mentees securing higher level leadership positions locally, nationally, or internationally. This was at all levels of seniority, but examples of mentees attaining senior leadership roles included full professor, head of department, or head of a clinical service. Importantly, mentees described learning from the mentoring process, and how this had transferred to their own teaching, supervision, and mentoring practices with others. This highlights a ‘ripple effect’ of mentoring across settings, geographical regions, and cultures.

## 4. Discussion

This paper presents my personal experience of mentoring, gained from supporting 605 mentees in public health, health education, and the health services over 30 years. This includes providing support for 231 mentees from 22 countries through one-to-one relationships (ranging from several months to 10+ years), and a further 374 mentees and interns supported through structured mentoring and internship programmes that I established and led. My main message from this combined experience is that mentoring is not only a privilege, but an essential activity leading to tangible outcomes for mentees and mentors alike. I have observed personal growth and development, career progression, and the attainment of notable success indicators (e.g., promotion, new job roles, prestigious awards, better work–life balance and improved wellbeing). I have myself experienced significant personal growth as well as recognition for the work and time involved. For example, the capacity-building and mentoring initiatives described here partially contributed to my own professional development and recognition through prestigious awards (such as my two Lord Dearing Awards) and my attainment of status as Principal Fellow of AdvanceHE [43]. AdvanceHE is a national organisation in the UK aiming to improve higher education for staff, students, and society. Principal Fellowship is the highest status awarded by the organisation and recognises a high level of impact. At the time of attainment, the evidence needed for Principal Fellowship included the following [43]: (i) successful, strategic leadership to enhance student learning, with a particular, but not necessarily exclusive, focus on enhancing teaching quality in institutional and/or (inter)national settings; (ii) establishing effective organisational policies and/or strategies for supporting and promoting others (e.g., through mentoring, coaching) with high-quality teaching and support for learning; (iii) championing, within institutional and/or wider settings, an integrated approach to academic practice (incorporating, for example, teaching, learning, research, scholarship, administration, etc.); and (iv) a sustained and successful commitment to, and engagement in, continuing professional development related to academic, institutional, and/or other professional practices. Some of the mentoring activities described in this paper provided evidence towards my own professional development goals in the pathway to AdvanceHE recognition. Learning and development from mentoring relationships is undoubtedly two-way.

In this paper, I described the diverse contexts in which I have provided mentoring, suggested ways in which each approach could be used (‘use cases’), and presented the benefits of each approach. It should be noted that I have not included in the 605 mentees the vast numbers of individuals I have supported through the routine academic practices of knowledge enhancement, one-off micro-mentoring activities, inductions, etc, but rather focused on those with whom I have developed more meaningful relationships through structured programmes or unstructured individual mentoring.

Mentoring is often considered within the context of two main philosophies: ‘developmental mentoring’ and ‘sponsorship mentoring’. Developmental mentoring focuses on the personal and professional growth of the mentee. In this way, the mentor acts as a guide, providing advice, feedback, and support to help the mentee develop their skills, knowledge, and confidence. The relationship is often long-term and centred around the mentee’s goals and aspirations. With sponsorship mentoring, the mentor actively advocates for the mentee, helping them gain visibility and access to opportunities within the organisation or industry. The mentor uses their influence and networks to promote the mentee’s career advancement. This type of mentoring is often more strategic and goal-oriented, focusing on achieving specific career milestones. Throughout my career, I have chosen not to focus solely on one approach but have instead chosen to be entirely adaptable based on the needs of individual mentees and the type of mentoring undertaken. While these are distinct approaches, both are critical for career advancement, and it is recognised that “mentors can be sponsors if highly placed and well connected” [44]. The process-oriented model of mentorship [31] includes passive functions, such as role modelling, in which a mentor acts as an example of the norms, attitudes, and behaviours required to achieve success [32,33]. In my view, role modelling, as a key function of mentoring, can apply to either developmental or sponsorship mentoring. It does not need to be confined to the requirements of a specific job role or position but can highlight the characteristics required for personal growth that have broader applicability, such as emotional intelligence, persistence, self-discipline, creativity, flexibility, curiosity, and a desire to improve.

Flexibility in mentoring style and approach is a key skill that will widen access to mentoring relationships for many individuals. According to Nuis and colleagues [3], “a good mentor is able to provide all types of support and will focus on offering particular types of support depending on the needs of the mentee”. There are occasions when a close match on specific characteristics can enhance a mentoring relationship through a deepened and shared understanding (e.g., matching by certain sociodemographic characteristics or specific professional groups). However, my experience of mentoring individuals at all career stages, from within and beyond my institution and country, from diverse cultural and professional backgrounds, and spanning higher education, industry, and the healthcare services, demonstrates the value of broadening mentoring ‘reach’ and offering rich, new perspectives to the relationship. Mentoring relationships that bridge academic, industry, and clinical spheres can provide mentees with a well-rounded perspective. Through these diverse experiences, mentees can gain a broader understanding of their field by seeing how theories, education, and research can be applied in real-world settings. They can develop a versatile skill set that is valuable across different sectors, thus increasing their future employability. It allows mentees to build a robust professional network that spans multiple areas, opening up more opportunities for collaboration and career advancement. Finally, receiving insights and advice from a mentor with a different background (profession or industry) can be incredibly enriching and inspiring.

The diversity of my mentees, and the contexts and settings in which I have provided mentoring, have instilled my belief that cross-discipline and cross-cultural mentoring serves to enhance versatility and adaptability, foster innovation and creativity, and enrich approaches to problem-solving. This requires continuous learning on the part of both the mentor and mentee but brings certain challenges. The key to success is maintaining an open dialogue about differences (e.g., characteristics, both cultural or professional). It is important to take time at the outset to understand the foundational concepts of specific fields, such as jargon, conceptual frameworks, professional expectations, and aspirations to which goals can be aligned. There are vast differences in work pace, decision-making, and hierarchy between fields and cultures which require open discussion to minimise cultural misunderstandings. There can be challenges of geographical location with international mentoring (e.g., only virtual meetings, differences in time zones), which can also manifest more locally with the changing nature of work in recent years, due to the sharp increase in flexible, hybrid, and remote working. However, such challenges may be overcome by levering technology, such as online meeting platforms and social media groups, either for individual or group support. Personally, I use technology to sustain mentoring relationships over time, to enable global reach, and to connect and network current mentees with alumni.

There are many theoretical frameworks that have informed and shaped my practices. I provide a Psychosocial Support System aligning with Kram’s theory of psychosocial functions [45]. For example, in one-to-one mentoring I engage in the affirmation of belonging (e.g., for individuals from under-represented groups), and provide support with navigating academic communities. Through my internship and structured mentoring programmes, I align with Tinto’s theory of social integration [46] by providing structures and processes that facilitate academic interests and connections with shared identities or visions.

My mentoring is a Learning Partnership, aligned with Learning Theories. For example, I advocate for adult learning (e.g., [47]) by facilitating mentees to be self-directed in their professional development and to reflect on past and current experiences, which supports confidence in their own capabilities. Viewed through a constructivist lens (e.g., [48]), my mentees accumulate new knowledge through experiences within their contexts and construct the meaning of those experiences through reflection to reconstruct new knowledge. In this way, I encourage mentees to critically reflect on their achievements and failures to construct new knowledge and awareness of their growing identities. I advocate for ‘action learning’ (learning by doing) [49], for example, my mentees have engaged in health promotion practice and interprofessional learning in industry settings through the WHIRL project [27]. Transformative learning [50] occurs when I establish communities of practice [51], for example, through the structured mentoring and internship programmes, and through my alumni and mentee networks.

My mentoring aligns with Developmental Theories. For example, through one-to-one mentoring I may adopt Levinson’s theory of adult development [52] in discussions relating to life transitions and career stages and how that relates to mentees adapting to new identities (e.g., transitioning through career stages, periods of illness, maternity leave, retirement planning). I consider Kram’s mentoring phases [38] as my mentoring relationships evolve over time, and the relationship needs to be redefined as mentees establish a new identity (e.g., during a transition to a new job or career, a move to a different country, a change in professional goals).

My mentoring aligns with Social Theories. I facilitate ‘socialisation’ through supporting skills and competency development to socialise mentees into their communities (e.g., healthcare, health educators). I facilitate the development of social capital [53] by engaging in role modelling (e.g., as a first-generation academic, to first-generation mentees), through the introduction of mentees to professional networks, and by continually reflecting on how mentees may be differentially evaluated based on characteristics such as socioeconomic status, country of birth or residence, race, ethnicity, gender identification, and disability.

My mentoring activity aligns with Organisational Learning Theory [54]. Giving two examples, first, in the WHIRL project [27], the structured clinical and peer mentoring helped mentees to develop competencies and ‘learn the ropes’ of health promotion practices, learning from their successes and mistakes to support their continued learning and development as a health professional and gain experience in industry settings. As a second example, through my internship programme, I provided a clear structure and processes for understanding both higher education (‘the institution’) and public health (‘the subject discipline’). Both types of mentoring brought shared visions and team learning, and changes in organisational culture, structure, and processes. These changes were associated not only with the existence of the mentoring programmes per se (for my own institution as the programme host), but also with the impacts and outcomes of the health innovations being delivered on the recipient organisations (healthcare and industry settings).

Finally, my work aligns with Career Support Theories. Specifically, I apply a range of career development functions as outlined by Kram [45]. I provide ‘sponsorship’ by visibly highlighting the skills, contributions, and achievements of my mentees. Where possible I ‘protect’ mentees from activities detrimental to their growth, such as diversions of focus or taking on too many activities. As a trained coach, I may utilise ‘coaching’ skills and techniques to support this process (noting that mentoring and coaching are different things, but the skills required may overlap). Where appropriate, I create or signpost to situations or projects that offer new ‘challenges’ for mentees.

Here, I advocate for the evaluation of mentoring through reflection, feedback, and formal surveys. By evaluating my own mentoring using validated measures, I provide a worked example of this approach. This worked example confirms that career and psychological support has been achieved through my mentoring relationships but has also led me to reflect on how such measures can be interpreted in different ways by individuals from diverse cultural backgrounds. This opens up opportunities for further exploration as to the validity of mentoring evaluation questionnaires in different settings, contexts, and populations. My evaluation identifies six key areas of benefit from mentoring relationships, including interpersonal relationships; networking opportunities; enhancing knowledge and skills; employment, reward, and recognition; support for under-served groups; and convening communities of practice.

We need to create a culture within higher education organisations that advocates for and supports mentoring more visibly. Johnson [55] boldly stated that “Unless institutional leaders in any college or university play a key role in (1) promoting a culture conducive to mentoring and (2) creating a structure to facilitate high rates of effective mentoring, the probability that any student or junior faculty member will be well mentored hovers somewhere between poor and unlikely”. Mentoring is a key process in ‘levelling the playing field’ for those experiencing barriers to progression. It is recognised that mentoring for young people experiencing socioeconomic disadvantages, in particular, is likely to have a more significant impact than for other groups [56]. In understanding the impact of my mentoring, I take a sociological stance based on Abelev [57]. As described by Wilson and colleagues [58], Abelev “draws heavily on Bourdieu’s concept of habitus defined as ‘…the norms, beliefs, speech patterns and interactional style that members of a group internalize and accept as doxa, or Truth, and then view as common sense, or the way things should be done’ (p. 135)” and argues that “one of the main areas in which the difference in habitus between socially advantaged and disadvantaged young people manifests itself is in their interaction with institutions” [56]. A mentor can therefore help to reduce this ‘interactional deficit’ by indirectly and directly ‘teaching’ mentees cultural literacy—a key factor required to advance, including knowledge of interactional styles, how to access resources, and a belief in one’s entitlement to resources within institutions.

My approach to mentoring has undoubtedly been influenced by that of my own mentors. I acted as a mentor from a very early stage of my career, but did not access formal mentoring support until I had reached a more senior position. I recently reflected on my experience of The Vice-Chancellor’s Mentoring Programme, which is part of my institution’s approach to developing an inclusive workplace culture by supporting senior leaders with protected characteristics. This structured mentoring programme was organised by the institution’s professional development team, and was funded by AdvanceHE as part of the Good Practice Grants (GPGs) which provide AdvanceHE members with the opportunity to develop and share their innovative practice across the higher education (HE) sector. Having secured a place on this programme in 2019, over the course of the year, I participated in a series of 1:1 mentoring meetings with my institution’s Vice-Chancellor, completed a Myers Briggs Type Indicator (MBTI) assessment [59], connected with other mentees on the programme as a group, and shadowed two senior university meetings. In my case, these meetings included a University Executive Board meeting, and a Partnerships and Stakeholder Relations Forum which focused on the University’s research and knowledge-exchange relationships with organisations in the ‘Major Corporates’ grouping. By 2024, the fifth year of the programme, 73 individuals had received mentoring support across the institution. Of these, the Vice-Chancellor personally mentored 36 senior leaders (of which I was one), with 57 unsuccessful applicants matched through the scheme to alternative Executive Board and other senior leader mentors. My personal reflections on this mentoring experience are published elsewhere as one of six ‘best practice’ case studies associated with this programme [60]. While this mentoring programme was beneficial to my own career development and included clear guidance on the expectations of mentees and mentors, it did not include supervision or oversight of my own mentoring practices with others (as this was not the focus of the programme). While I personally learned from exposure to the mentoring practices of the Vice-Chancellor and modelled these good practices with my own mentees, it is not known whether other senior leaders undertaking the programme did the same, and what benefits may have occurred (e.g., the ‘ripple effects’). Future structured mentoring programmes might benefit from the inclusion of mentoring supervision for both parties, and some exploration of whether such programmes can enhance the ‘mobility’ of mentoring practices throughout higher education to explore the transferability of good mentoring practices across the sector, through observation, sharing, modelling, and implementing them with diverse mentees in different contexts and settings.

## 5. Conclusions

Overall, my mentoring experiences over 30 years highlight the benefits of structured and unstructured mentoring for individual career development, and how this development has subsequently impacted public health, health education, and/or the health services. The key lessons I have learned from my mentoring experiences are shared in Figure 7. By reflecting on my own experiences, key questions have been answered which could inform the mentoring decisions of others. These are: (i) What are the specific advantages that mentoring offers to faculty members in terms of professional development, career advancement, and personal growth? (ii) In what contexts and environments could mentoring take place? (iii) Which traits or qualities are important for mentors and mentees to ensure productive and supportive mentoring relationships? (iv) What are the key challenges in mentoring and what prevention strategies should be adopted? (v) How can theoretical frameworks guide the implementation and practice of mentoring in higher education?

With the significant benefits of mentoring to the workforce, higher education organisations, and society which are outlined here and widely acknowledged [61], this paper is a ‘call to action’ for academic faculty to engage with mentoring which enriches professional life and should be an integral part of an academic’s role. Organisations should recognise and advocate for the value of mentoring to individuals and institutions, and protected time and training to facilitate mentoring should be provided for both mentors and mentees [22,62].

## Figures and Tables

**Figure 1 ijerph-22-00729-f001:**
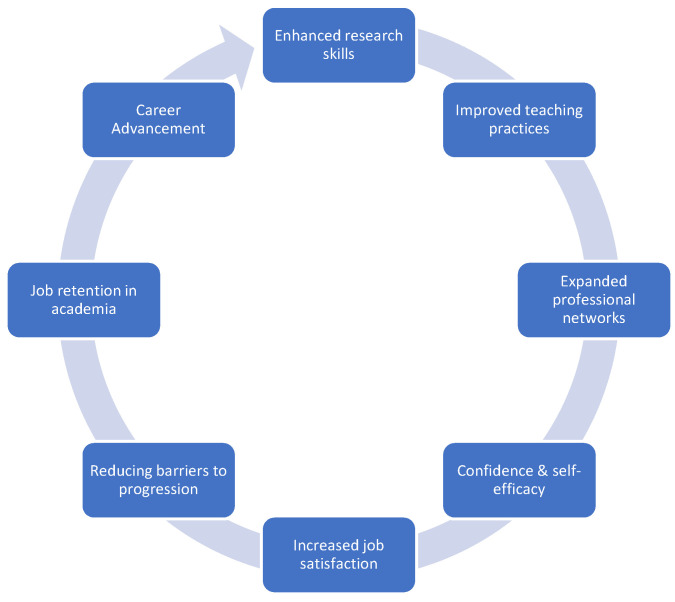
Impact of mentoring on academic careers.

**Figure 2 ijerph-22-00729-f002:**
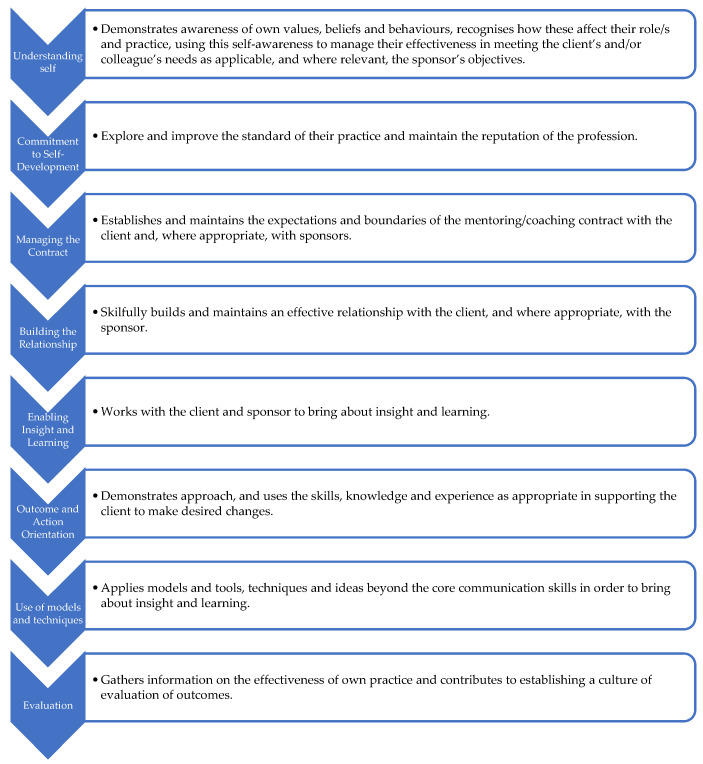
EMCC Global eight core standards for mentoring, coaching and leadership.

**Figure 3 ijerph-22-00729-f003:**
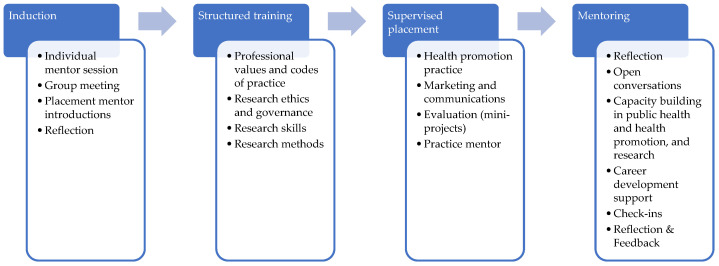
Mentoring within an interprofessional internship.

**Figure 4 ijerph-22-00729-f004:**
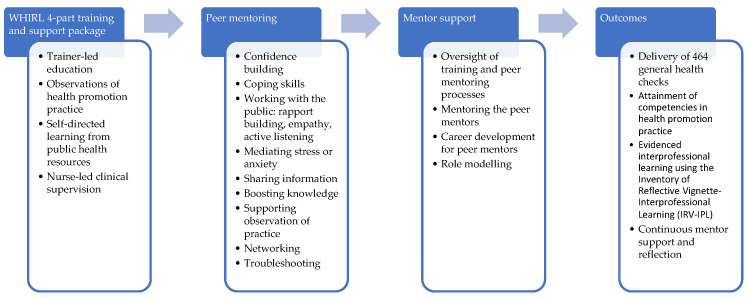
Mentoring within the WHIRL interprofessional learning programme in Test@Work.

**Figure 5 ijerph-22-00729-f005:**
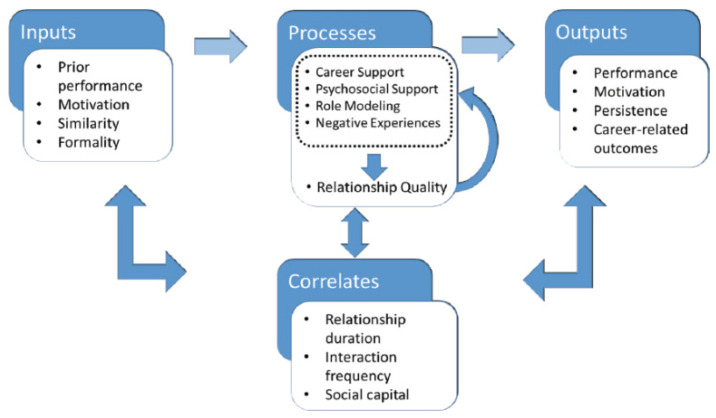
Simplified process-oriented model of mentorship. Source: [32], adapted from [31].

**Figure 6 ijerph-22-00729-f006:**
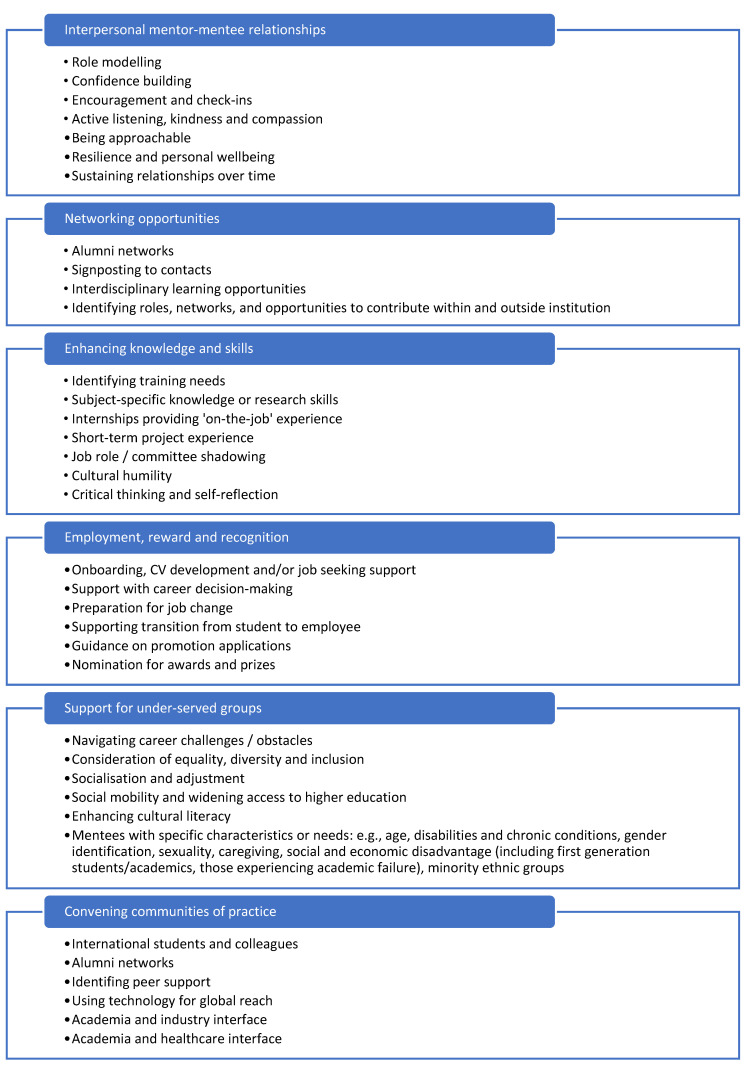
Key areas of focus for a positive mentoring experience.

**Figure 7 ijerph-22-00729-f007:**
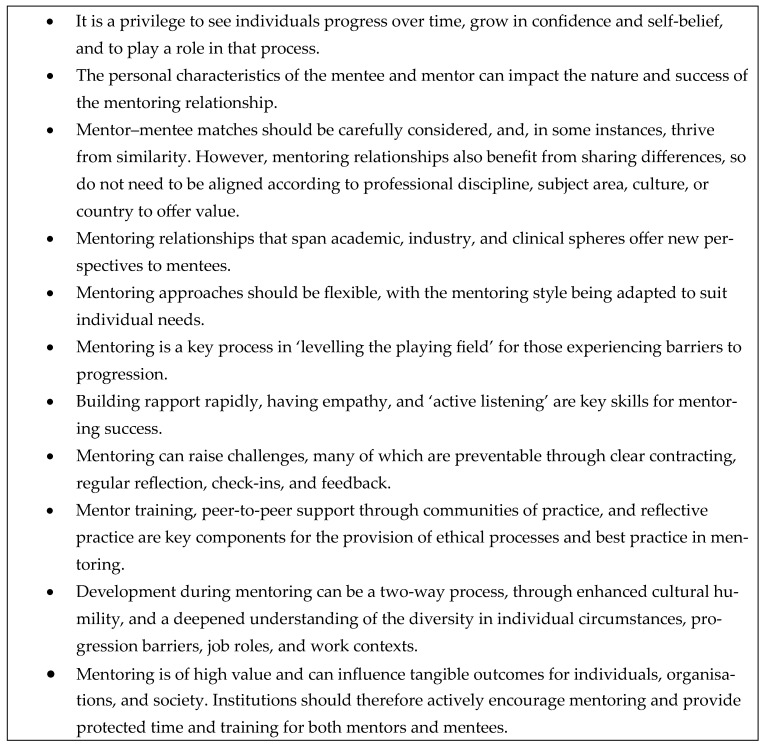
Key messages.

**Table 1 ijerph-22-00729-t001:** Mentoring types, use cases, and benefits.

Mentoring Type	Use Cases	Benefits
Micro-mentoring	Short-term or single session Focused on a topic, skill, or short-term project Shadowing opportunities Support with awards and recognition (e.g., reviewing promotion applications, supporting development of applications for achievement awards) Individual approach	Low time commitment Short-term/measurable objectives Enhances organisational understanding Improves academic performance Supports achievement and productivity
Inducting new staff	Onboarding Short-term Focused on job role profile, teams, and environment Individual approach	Low time commitment Short-term/measurable objectives Rapport building Socialisation and adjustment
Peer mentoring	Inclusion within a team Environment and team support, and specific skills or characteristics (e.g., disability) Individual approach	Relationship building Signposting Socialisation and adjustment
Career transition moments	Researcher to tenured academic Student to early career researcher Job exchange (e.g., new comparable research contract or clinical role; move from academia to industry or reverse) Individual or group approach	Low/medium time commitment Short-term/measurable objectives Eases transitions Preparation for challenging experiences Individual enablement
Career advancement mentoring	Professional development Individual approach	Individual enablement Succession planning Future-proofing Staff retention
Diversity mentoring	Inclusion (protected characteristics) Career development Leadership development Individual or group approach	Support for under-served individuals Improvement of diversity in leadership or specific job roles Reduction in unconscious bias Builds social capital and social mobility Enhances cultural humility Widens access to higher education Facilitates adjustment and achievement
Knowledge sharing mentoring	Upskilling Training Career development Individual or group approach	Building skills in specific areas, functions, or use of technology Engages senior/experienced staff in skill or knowledge sharing
Collaborative learning and support mentoring	Career development Upskilling Group approach	Rapport building Intraprofessional relationship building and support
Leadership development mentoring	Upskilling High-potential development Individual or group approach	Enhances leadership potential

**Table 2 ijerph-22-00729-t002:** Planning and running a structured mentoring programme.

Stage	Considerations
Planning	Establish programme purpose and goals Clear programme leadership, and senior leadership ‘buy-in’ Identify resources (e.g., training materials for mentor training, support staff) Establish clear guidelines, expectations, and organisational goals Focus on target groups (e.g., staffing groups, under-represented groups) Decide on the nature of mentoring (e.g., one-to-one mentoring relationships, peer mentoring, group mentoring, or reverse mentoring) Identify suitably experienced mentors Identify prospective mentees Consider how participants enrol Determine mentor/mentee matching process Decide on programme duration Establish feedback mechanisms Consider offering awards/rewards/benefits for participation Advertise the programme by highlighting key benefits and salient features
Initiation	Introduce programme organisers Deliver training to mentors and mentees Emphasise the three C’s of mentorship—communication, clarity, and commitment Perform careful matching of mentors with mentees (e.g., consider sociodemographic characteristics, education, experience, subject expertise, location) Establish clear communication channels Ensure expectations are realistic Dyads set up meetings Create plan for tracking progress
Mentoring	Identify areas for growth and development Flexibility with frequency of meetings and mode of communication Regular feedback Consider recommending formal training and professional development
Evaluation	Assess mentee progress through appropriate method (e.g., reflection, discussion, goal attainment) Collect mentor/programme feedback (e.g., reflection, discussion, evaluation survey, focus group)

**Table 3 ijerph-22-00729-t003:** Challenges in mentoring and suggested actions.

Challenges	Actions
Mismatched expectations	Discuss expectations up front Consider providing written guidance Build shared understanding Suggest alternative pairing if relevant
Lack of communication of stressors by mentees	Reinforce confidentiality Build rapport and trust Provide multiple contact options Provide open feedback
Lack of goal setting by mentees	Discuss expectations up front Consider providing written guidance Assist with setting specific and measurable goals Focus on goals as meeting discussion points
Lack of planning by mentees	Discuss expectations up front Send reminders prior to meeting Emphasise individual accountability
Lack of mentor training	Undertake mentor training Seek diverse mentoring experiences
Balancing time and commitment	Provide multiple contact options Provide options for remote contacts Set clear limitations and boundaries Adhere to agreed contact frequency Adjust number of mentees if necessary
Overcoming resistance to change	Build mentee confidence Encourage mentee to set realistic goals Appreciate mentees’ personal and work situation and competing demands
Maintaining mentee engagement	Evaluate outcomes to support reflection Regularly assess mentoring needs Adapt mentoring style to suit the relationship Offer flexibility where possible Signpost to an alternative pairing if relevant
Dealing with mentee inexperience	Identify knowledge/skills gaps Provide opportunities to learn/develop skills (where appropriate and without abuse of power) Signpost to opportunities
Fostering mentee independence	Build mentee confidence Set clear limitations and boundaries Realistic goal setting Encourage mentee decision-making Enable mentoring closure
Unknown outcomes	Measure outcomes through the following: 360° feedback for mentor Informal evaluation: regular reflection/discussion Formal evaluation: gather metrics

**Table 4 ijerph-22-00729-t004:** Career and Psychosocial Support Scales.

Career Support Scale (*n* = 103)	Min–Max *	Mean	S.D.
Provides you with opportunities that stretch you professionally	5–7	6.80	0.492
Creates opportunities for visibility for you in your career	4–7	6.76	0.551
Opens doors for you professionally	5–7	6.82	0.437
Acts as a sponsor for you	2–7	6.84	0.622
Acts as a buffer for you from situations that could threaten your career achievement	1–7	6.48	1.092
			Total (33.66/35)
**Psychosocial Support Scale (*n* = 103)**	Min–Max *	Mean	S.D.
Cares and shares in ways that extend beyond the requirements of work	4–7	6.89	0.441
Counsels you on non-work-related issues	4–7	6.72	0.663
Offers you support, respect, and encouragement	7–7	7.00	0
Is a friend of yours	3–7	6.25	0.957
Confirms and affirms your identify and sense of self	5–7	6.95	0.257
			Total (33.81/35)

* Response scale ranged from 1 (never, not at all) to 7 (to the maximum extent possible).

## Data Availability

The original contributions presented in this study are included in the article. Further inquiries can be directed to the corresponding author.

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
