# Peer review of "Workforce Career Development in Public Health, Health Education, and the Health Services: Insights from 30 Years of Cross-Disciplinary National and International Mentoring"

_ijerph, 2025, doi:10.3390/ijerph22050729_

Round 1
Reviewer 1 Report
Comments and Suggestions for Authors
I read your piece with fascination and admiration. You appear to have been in thoughtfully reflective mentoring relationships across a very meaningful career. Kudos! That said, what I struggled with here was what you see as your contribution to a relevant scholarship or literature(s) here? You have mentored in multiple (and quite different contexts (e.g. PhD students, tenure track academics, internships and so on) and surely these wildly different fora yielded specific fresh insights to those literature(s) that you could highlight? Likewise, while many Northern university faculty members mentor students from many nations, you appear to have done so not only in the academy but in other contexts as well. What did you learn from that diverse experience that has not been shared in relevant literature? Perhaps you could address these queries fairly quickly by showing how your cumulative experiences can be shown to advance one (or more) of the forms of mentoring with which you have engaged?
Reviewer 2 Report
Comments and Suggestions for Authors
This is an unusual paper in that it is a self-reflection based on personal experience.The author navigates well the complexity of balancing self-bias and scientific enquiry. I find many similarities with the kind of "semi-detached" analysis expected in a professional doctorate. Its strengths include the bveradth of mentoring situations analysed and compared. I have no hesitation in recommending publication.
My only suggestion is to make clear the fact that there are two philosophies of mentoring represented in the academic literature. One conflates mentoring and sponsorship; the other does not. The author refigures to both of these but does not indicate or discuss this significant difference. A short paragraph exploring her experience in this context would be useful.
Reviewer 3 Report
Comments and Suggestions for Authors
I think the paper is really good and well written and referenced. In HE there is emphasis on mentoring and often it is associated to induction, supporting ERCs, supporting HEA fellowship achievement or other qualifications being gained and promotion. The training for mentors in HE varies from none, a powerpoint to read, a workshop to attend or masters level short course. I wonder whether more could be talked about in the paper about what training mentors might need. This has implications for the quality of the mentoring and mentoring outcomes. I am also mindful that mentoring in HE largely ignores the ethical matter of having supervision for mentoring practice as advocated by the EMCC. When I have trained mentors using a masters level short course, I have encouraged HE staff to set up a community of practice where they undertake peer-supervision and I have also offered colleagues supervision for reflecting on practice. I felt that this ethical and best practice matter was somewhat missing from the paper. Did you receive mentoring supervision for your own extensive practice and if not, how might have this helped? You do talk about being mentored, did any of this link to you being mentored on your mentoring? A final thought from me was what impact does all of this have on public health? Typing the paper back to the scope of the journal would be helpful in findings and conclusion. I imagine that staff being mentored have been better in their healthcare practice? Possibly have had better results for patients as a result? Have found groundbreaking solutions to health problems in their research. Provide better healthcare education to students?
Round 2
Reviewer 1 Report
Comments and Suggestions for Authors
Thank you for this revision, which I read with interest. I do want to suggest,having read this iteration, that you carefully edit your revised abstract and MS with an eye to eliminating your use of passive voice within it. As you know, passive voice effectively eliminates a subject from affected sentences, leaving readers to guess the same.
Author Response
Dear Reviewer,
Thank you for taking the time to review the second version of my manuscript. I have endeavoured to make changes through the abstract and the whole manuscript in response to your comment. Active voice has been used where relevant.
Many thanks,
Prof. Holly Blake